

**Reactive mercury flux measurements using cation exchange membranes**
Matthieu B. Miller[1*], Mae S. Gustin[2], Grant C. Edwards[1,3]
[1]Faculty of Science and Engineering, Department of Environmental Science, Macquarie University, Sydney, NSW,
2113, Australia
[2]Department of Natural Resources and Environmental Science, University of Nevada, Reno NV, 89557, United
States
[3]Deceased 10 September 2018.
*Correspondence to*: Mae Sexauer Gustin mgustin@cabnr.unr.edu
**ABSTRACT**
A method was developed to measure gaseous oxidized mercury (GOM) air-surface exchange
using 2 replicated dynamic flux chambers (DFCs) in conjunction with cation exchange
membrane (CEM) filters. The experimental design and method was developed and tested in a
laboratory setting, using materials collected from industrial scale open pit gold mines in central
Nevada, USA. Materials used included waste rock, heap leach ore, and tailings, with substrate
concentrations ranging from 0.1 to 40 μg g$^{-1}$ total mercury (THg). CEM filters were used to
capture GOM from the DFC sample lines while a Tekran® 2537A analyzer measured GEM
concurrently. Previous and ongoing work demonstrated that CEM do not collect GEM and
efficiently collects multiple compounds of GOM. Positive GOM emission rates up to 4000 pg m$^{-}$
$^2$ h$^{-1}$ were measured from tailings materials with high Hg substrate concentrations, and this has
significant implication with respect to air-Hg surface exchange. GOM flux was variable for
lower Hg concentration substrates, with both emission and deposition observed, and this
wasaffected by ambient air GOM concentrations. For substrates that experienced GOM
deposition, deposition velocities were in the range 0.01 – 0.07 cm s$^{-1}$.



# 1   Introduction

On 16 August 2017, the United Nations Environment Program (UNEP) Minamata Convention
came into force with a mission to protect human health and the environment from exposure to
the toxic effects of mercury (Hg) and its various compounds (UNEP, 2013). The Convention
proposes to fulfill this mission through a strategy of globally coordinated scientific research, and
ongoing monitoring of Hg in the environment where possible. The mission poses a significant
challenge as there are over 3000 known Hg contaminated sites worldwide due to mining and
industrial activity, not including legacy and small scale operations involving Hg (Kocman et al.,
2013; Krabbenhoft and Sunderland, 2013). Past emissions were dominantly from natural sources,
but are now largely from anthropogenic activities and have led to large reservoirs of Hg in all
environmental compartments (Pacyna et al., 2016; Pirrone et al., 2010; Streets et al., 2017). Even
in the absence of new anthropogenic emissions, cycling of Hg between different spheres will
continue at large scales and in possibly unexpected directions, driven by processes and
mechanisms that are not fully understood, and susceptible to increasing human and climate
perturbations (Obrist et al., 2018).
The atmosphere, a global common, is the dominant conduit for transport, transformation,
emission, deposition, and re-emission of Hg and Hg compounds amongst Earth's ecosystems.
Mercury in the atmosphere is classified on the basis of three physiochemical forms: gaseous
elemental mercury (GEM), gaseous oxidized mercury (GOM), and particulate bound mercury
(PBM), with PBM and GOM defined together as reactive mercury (RM = GOM + PBM)(Gustin
et al., 2015). Atmospheric concentrations of GEM can be measured with well calibrated



analytical instruments, whereas the quantification of GOM and PBM has depended on
operationally defined methods with demonstrably large uncertainty (Cheng and Zhang, 2017;
Gustin et al., 2015; Gustin et al., 2013; Jaffe et al., 2014; Zhang et al., 2017).
Scientific understanding of the biogeochemical cycling of Hg is currently inadequate due to the
large uncertainties in measurements, and general difficulty of research on atmospheric Hg
chemistry (Jaffe et al., 2014). Atmospheric GEM is relatively inert, has a low deposition rate,
and hence a relatively long atmospheric lifetime ranging from minutes to 1 year (Krabbenhoft
and Sunderland, 2013; Zhang et al., 2009; Gustin et al., 2013). Observations of rapid depletion
of GEM from the atmosphere suggests that deposition and re-emission on short time scales is an
important process driving movement (Howard and Edwards, 2018; Lu et al., 2001; Schroeder et
al., 1998). In contrast to GEM, GOM compounds have higher dry deposition velocities (Zhang et
al., 2009). Knowledge of concentrations, chemistry, and processes forming atmospheric GOM is
critical for understanding how Hg moves and impacts ecosystems globally. Recent research on
GOM has demonstrated that compounds in air vary both spatially and temporally to a
considerable extent, pointing to the need for extensive measurements of concentrations and
identifying the specific compounds (Gustin et al., 2016; Huang and Gustin, 2015b; Huang et al.,
2014; Huang et al., 2017).
Fluxes of total gaseous mercury (TGM) and GEM have been successfully measured in many
environments, providing insights into air-surface exchange that is critical for understanding
biogeochemical cycling of GEM (Yannick et al., 2016; Zhu et al., 2016). In contrast, there are
few direct measurements of GOM air-surface exchange (Zhang et al., 2009). GOM fluxes
reported in the literature are largely based on measurements made with the KCl denuder as the





GOM collection method, as well as a passive Hg dry deposition sampler (Brooks et al., 2008;
Castro et al., 2012; Engle et al., 2005; Lindberg et al., 2002; Lindberg and Stratton, 1998; Lyman
et al., 2009; Lyman et al., 2007; Malcolm and Keeler, 2002; Poissant et al., 2004; Rea et al.,
2000; Rothenberg et al., 2010; Sather et al., 2013; Skov et al., 2006; Zhang et al., 2005; Huang et
al., 2015b). The few direct GOM-specific air-surface exchange measurements using KCl
denuder-based approaches have been undertaken with mixed results (Brooks et al., 2008; Skov et
al., 2006). Moreover, recent evaluation of KCl denuder approaches show these methods tend to
underestimate GOM and are subject to interferences due to ozone and water vapor (Huang and
Gustin, 2015a; Lyman et al., 2010; McClure et al., 2014). Understanding GOM air-surface
exchange processes would facilitate the Hg scientific community's ability to move forward in
understanding the biogeochemical cycling of atmospheric Hg.
In this study, we developed and applied a novel experimental approach for direct measurement
of GOM fluxes using cation exchange membrane (CEM) filters. CEM filters have been
successfully deployed to measure GOM in ambient air in previous studies (Gustin et al., 2016;
Huang et al., 2017; Huang et al., 2013; Marusczak et al., 2017; Pierce and Gustin, 2017), and
their use here was modified to allow for the determination of GOM air-surface exchange.  A
recent study (Miller et al., 2018) and ongoing work (unpublished data) have demonstrated that
GEM is not collected by the CEM and it efficiently collects a variety of GOM compounds. Here
is described a series of experiments that were conducted to optimize the CEM/GOM flux
methodology, and then, the resulting method was subsequently applied to determine fluxes over
both background waste rock material and Hg enriched mining materials.  Our research
hypothesis was the system developed would provide a means of determining if GOM is



deposited or emitted from substrates, and this system would allow us to develop an
understanding of GOM air-surface exchange.
**Methods**

**2.1 Materials**

Substrate materials used for measuring Hg flux were acquired from industrial scale, open pit
gold mines in central Nevada, and include waste rock, heap leach ore, and tailings.  Materials
were obtained from 4 ongoing mining operations: Twin Creeks (TC) and Lone Tree (LT)
operated by Newmont Mining Corporation, and Cortez Pipeline (CP) and Gold Strike (GS)
operated by Barrick Gold Corporation (Fig. 1; Table 1). There were 3 types of materials: waste
rock called cap (C), heap leach material (L), and tailings (T or tails).  (Waste rock is non-
mineralized low Hg overburden consisting of alluvium or hard rock that covered the ore body
prior to mining. For this study, materials were sampled from the waste rock piles specifically set
aside for future capping and site reclamation, and as such is referred to as cap material. Heap
leach is low-grade ore blasted from the mine wall and "heaped" on an impoundment for
irrigation with dilute cyanide leach solution for gold extraction. Tailings are the waste remnant
of high-grade ore that has been pulverized by mechanical ball milling and undergone
thermal/chemical treatment to extract gold. All mining substrate materials were collected in
September 2010, for measurements of GEM flux under controlled conditions (Miller and Gustin,

2013).

Each material was divided into replicate trays (50 x 50 x 7 cm$^3$ plywood lined with 3mil
polyvinyl sheet) and stored inside a greenhouse bay at the University of Nevada Reno



Agricultural Experiment Complex. At the onset of the GOM flux experiments described in this
paper, substrates were undisturbed for ~3 years, and were completely dry and well
compacted/consolidated from previous watering experiments. A circular chamber footprint
impression previously existed from prior experiments provided an excellent existing contact for
the chamber base. Comparing GEM flux measurements made in this study with the previous
measurements, an overall trend of decreasing GEM flux over time was observed. This fits a
hypothesis suggested by Eckley et al. (2011b) of a long term reduction in GEM evasion from a
mine substrate as time from disturbance increases.
**2.2 Methodology**
GOM flux was measured by modifying our existing GEM flux method consisting of a dynamic
flux chamber (DFC) and an ambient air Hg analyzer (Tekran® 2537A) after the methods of
Eckley et al. (2010; 2011a) and Miller et al. (2011).
Briefly, the GEM flux system used a 2537A in conjunction with a cylindrical DFC (footprint
0.036 $m^2$) made of molded Teflon film (0.19 mm thick) over a rigid Teflon frame (1.5 mm
thickness), with a total internal volume of 2.0 Liters (Fig. 2). Air enters the chamber through 24
inlet holes (1 cm diameter) spaced 2.5 cm apart around the perimeter and 2.0 cm above the
bottom edge. The sample outlet is 0.625 cm diameter PTFE tubing at the top-center of the
chamber, and sample inlet air is measured through equivalent tubing at the height of the chamber
inlet holes. A Tekran® Automated Dual Switching (TADS) unit was used to cycle the sample
flow (1.0 Lpm) between the chamber inlet and outlet lines in sequential 10 min intervals (two 5
min samples on each line). GEM in each 5 min sample volume was quantified automatically by
the 2537A analyzer using pre-concentration on gold traps followed by thermal desorption (500





°C) and cold vapor atomic fluorescence spectrometry (CVAFS). The difference in Hg
concentration between the outlet and inlet air ($C_o$ – $C_i$) referred to as $\Delta C$, and was used to
calculate flux by Eq. 1:
$$F = Q * (C_o - C_i)/A \quad (1)$$
where F is the net Hg flux (ng m$^{-2}$ h$^{-1}$), Q is the flow rate through the chamber (m$^3$ h$^{-1}$), $C_o$ is the
mean concentration of two consecutive 5 min outlet air samples (ng m$^{-3}$), $C_i$ is the mean
concentration of inlet air in the samples before and after $C_o$, and A is the area of the substrate
under the chamber (m$^2$). The sign of $\Delta C$ indicates the direction of flux with positive being
emission and negative being deposition.
The two modified systems included fitting the DFC inlet and outlet sample lines with 2-stage
disc filter assemblies (Savillex$^©$ 47 mm PFA Teflon Filter Holder) holding two inline
polysulfone cation exchange membranes (CEM 0.8 μm, Mustang$^®$ S, Pall Corporation) (Fig. 2).
CEMs preferentially capture GOM compounds while allowing GEM to pass freely (Miller et al.
2018; unpublished data). The first upstream CEM served as the primary collection filter, while
the second downstream CEM captured any Hg escaping from or missed by the first filter
(referred to as breakthrough). With CEM filters deployed at the front of the sample lines, in
conjunction with the 0.2 μm particulate filter at the rear sample inlet of the Tekran$^®$, GOM was
scrubbed from the sample flow and all Hg measured downstream on the 2537A is in GEM form.
An additional set of CEM filters was deployed simultaneously, using sample pumps and mass
flow controllers set to match the 2537A sample flow rate (1.0 Lpm, Fig. 2) at a height of 2 m



above the flux system. This was done to measure background ambient GOM concentrations in
the greenhouse.
With both the Tekran® and external pump controlling CEM sample lines at 1 Lpm, total flow
through the DFC was 2.0 Lpm that provides a chamber turn over time (TOT) of 1 minute. At this
flow rate, flow velocity through the chamber is laminar with no turbulent eddy formation, as
determined through computational fluid dynamic (CFD) modelling of the chamber geometry
performed by Eckley et al. (2010). Thus the chamber design of Eckley et al. (2010) is optimal in
terms of not disturbing the substrate being studied. With low velocity, non-turbulent flow,
particle entrainment from the substrate surfaces is not expected, especially for particle sizes
greater than the CEM filter pore size of 0.8 μm.
**2.3 Analyses**
After flux measurements, CEM filters were collected into sterile 50 mL polypropylene centrifuge
tubes, and frozen at -20 °C until analyses (within 14 days of collection). Filters were analyzed for
total Hg by aqueous digestion and cold vapor atomic fluorescence spectrometry (CVAFS, EPA
Method 1631, Rev. E) using a Tekran 2600 system, with total Hg operationally equivalent to
total GOM. The blank Hg mass that can be expected on an unused CEM filter (median = 68 pg, n
= 56) was determined from clean filters collected with every set of measurements, and the
median blank value was subtracted from all sample values. Breakthrough is defined as the
amount of Hg on the secondary filter as a percent of the total Hg collected on both filters (after
blank correction), and overall median breakthrough was low (4.2%, n = 222).





The GOM concentration in the inlet/outlet sample air was calculated by combining the blank-
corrected primary and secondary filters into a total Hg mass per sample line and dividing by the
respective sample air volume.  The difference in GOM concentration between the inlet and outlet
lines provided a $\Delta C_{GOM}$, with this multiplied by sample flow providing the GOM emission rate
(pg h$^{-1}$). Reactive Hg flux (pg m$^{-2}$ h$^{-1}$) was calculated with Eq. 1 using $\Delta C_{GOM}$ values, the flow
rate (1.0 ± 0.005 Lpm) and the chamber footprint (0.036 m$^2$).
The GOM flux detection limit was determined by the minimum statistically resolvable difference
between $C_o$ and $C_i$, i.e. the smallest meaningful $\Delta C_{GOM}$ that could be measured (Fig. 3). The $C_o$
and $C_i$ concentrations were based on two measurements: total Hg on the CEM filters as
determined by analysis on the Tekran® 2600 system, and total sample volume as determined by
the mass flow-controlled sample rate and time. The MFC precision was ± 0.5% (± 0.005 Lpm at
1.0 Lpm) and the detection limit of the 2600 was 1 ppt, or ~ 53 pg per reagent blank in a clean 50
mL collection tube. The median mass of Hg on the blank CEM filters was 68 pg and was used as
the practical detection limit for the method. As the distribution of CEM blank values was non-
normal and skewed heavily to the right (Fig. 3a), the 95% confidence interval around the median
(58 – 73 pg) was used to define a minimum detectable GOM concentration (Fig. 3b). For
example, in a 24 h sample, the minimum detectable GOM concentration would be ~ 3.5 pg m$^{-3}$ to
exceed the upper 95% confidence limit of 73 pg. In a 24 h sample the minimum resolvable
$\Delta C_{GOM}$ would be 13.5 pg m$^{-3}$ (sum of upper + lower confidence intervals, + 3 pg m$^{-3}$ to account
for flow precision, converted to 24 h sample volume concentration). This was taken as the upper
minimum $\Delta C_{GOM}$ value.





Chamber blanks were determined for each Teflon DFC (referred to as Chamber A and Chamber
B) by measuring flux over a clean Teflon sheet. Chamber blank emission rates were measured
immediately following chamber cleanings (24 h acid wash, 10% HNO₃), and then between
substrate types (i.e. cap, leach, tailings). For the summer measurement period, the median
chamber blank $\Delta C_{GOM}$ for both Chambers A and B (A = 13 pg m⁻³, B = 14 pg m⁻³, n = 12) was at
the detection limit, so chamber blanks were not subtracted from material fluxes. This was due to
high ambient GOM concentrations in the greenhouse. For the following winter measurement
period, the median chamber blank $\Delta C_{GOM}$ for both chambers was significantly negative (A = -60
pg m⁻³, B= -60 pg m⁻³, n = 6), and the median chamber blank was -215 pg m⁻² h⁻¹, indicating that
the chamber and the blank Teflon sheet were acting as depositional surfaces for GOM. For these
measurements, the chamber blanks were subtracted. For materials that demonstrated net GOM
deposition, the deposition velocity ($V_d$ cm s⁻¹) was calculated using Eq. 2:
$$V_d \; = \; \text{flux} \; (ng \; m^{-2} \; h^{-1}) \; / \; \text{air concentration} \; (ng \; m^{-3}) \; * \; (100/3600) \quad (2)$$
All fluxes were measured in the University of Nevada-Reno Agricultural Experiment Station
greenhouse. Meteorological parameters were measured synchronously with flux and recorded in
5 min averages, including temperature and relative humidity (HMP45C, Campbell Scientific®),
substrate temperature (C107, Campbell Scientific®), and solar radiation (LI-200X, LiCor®).
Rudimentary climate control was provided by ventilation fans pulling outside air across the
greenhouse bay from intakes on the opposite side.  Fluxes were measured on the "upwind" side
of the bay in a variety of orientations (see below).
Data was processed in Microsoft Excel (version 16.22) and RStudio® (version 3.2.2).



**2.4 Development of the method**
Two Tekran® 2537A analyzers with associated GOM filter systems were used to simultaneously
measure flux from two replicate trays (one for each A and B system) of each sample material.
The duplicated systems allowed a total of four GOM flux measurements to be collected each
time a material was tested, two from the Tekran® sample lines and two from the external pump
lines. The intention of the duplicate systems was to evaluate consistency and repeatability of the
measurements. However, during the initial method testing it became apparent that the position of
the trays and the inlet sample lines was an important variable. For example, strong GOM
emission from an "upwind" tray could substantially increase the inlet GOM concentration of the
"downwind" tray, resulting in a negative $\Delta C_{GOM}$ and apparent deposition. Such contradictory
results compelled us to test a variety of orientations for the two systems. The best results were
achieved in the final design by placing both trays all the way against the intake wall of the
greenhouse bay, and separating them laterally by 1.5 m, with all equipment located downwind.
This configuration provided the most uniform inlet air concentrations for both systems.
**2.5 Limitations of method**
The use of filter membranes in conjunction with a DFC to measure GOM flux has several
limitations.  Given the low concentrations of GOM, a relatively long sampling time of at least 24
h is required to capture a sufficient mass for quantification on low Hg containing substrates.
Despite the limitations, this method moves us a step further for understanding GOM flux. The
low temporal resolution limits analysis of the factors controlling GOM flux, since there will be
changes on a diel cycle similar to observations of GEM flux (c.f. Gustin et al., 2013).  In



237 addition, flow rate and surface area could be increased, but in order to change this method a

238 number of proof of concept tests would have to be done.

239 A necessary condition of the DFC method is the placement of flux chambers directly on a

240 material, resulting in artificial modification of surface conditions and a presumptive influence on

241 the magnitude of flux.  In our study, this limitation is immaterial, as the entire experimental setup

242 constitutes an artificial environment, and we are more interested in 1) our ability to

243 experimentally detect measurable GOM flux, and 2) the qualitative direction of flux versus

244 absolute quantification. As the method develops, it will become necessary to more thoroughly

245 evaluate the effects of the DFC on GOM flux.

246 Lastly, CEM were analyzed within two weeks of collection. Pierce et al. (2017) demonstrated

247 that little $HgCl_2$ was lost from the membranes over this time period. This needs to be tested for

248 other GOM compounds thought to be present in the atmosphere (Huang et al., 2013; Gustin et

249 al., 2016).

250 **3 Results**

251 **3.1 GOM flux measurement repeatability**

252 The first test of the optimized configuration was a set of replicate measurements made on a

253 single material, to assess method repeatability. The material used for this test was a heap leach

254 ore of intermediate total Hg concentration (TCL, $13.2 \pm 2.0$ μg g$^{-1}$).  Filters were deployed in

255 three consecutive sample sets of approximately 72 h each, for a total of 12 replicate filter flux

256 measurements (i.e. two GOM flux measurements on two separate replicate trays, three times).

257 The mean GEM flux from all samples was $167 \pm 57$ ng m$^{-2}$ h$^{-1}$ (n = 6). The duplicate



measurements of background greenhouse GOM showed a mean concentration of $13 \pm 7$ pg m$^{-3}$
(n = 12). The $\Delta C_{GOM}$ was well above detection for all measurements (median 97 pg m$^{-3}$, n = 12),
and the mean GOM flux over all 12 samples was $370 \pm 80$ pg m$^{-2}$ h$^{-1}$ (n = 12) and ranged from
240 to 490 pg m$^{-2}$ h$^{-1}$ (Fig. 4).
The mean relative percent difference (RPD) of GOM flux measured between Pump and Tekran
sample lines on the same tray was $6.5 \pm 3.9\%$, and mean RPD between trays was 21.4%. These
replicate measurements of a single material consistently showed the same direction and
magnitude of GOM flux with good agreement.
3.2 **GOM flux measurement replication over expanded range of materials: Summer**
Following the triplicate measurement of TCL (May), follow up testing was conducted on a series
of three additional materials (July – August): a low to intermediate Hg cap material and heap
leach ore (TCC 0.2 µg g$^{-1}$, LTL 0.6 µg g$^{-1}$), and a high Hg tailings (TCT 36 µg g$^{-1}$).
Measurement time was reduced to 48 h for this series of materials, as the previous deployments
had total GOM loading well above detection. However, several of the LTL and TCC flux
measurements resulted in insignificant $\Delta C_{GOM}$ values (Fig. 5). In these cases, GOM flux could
not be discriminated, and these values were excluded from subsequent analysis.
A comparison of GOM flux measured by the replicate sample lines on each system (Tekran®
sample flow and external pump sample flow) show a relationship of 1:1 (Fig. 6) that indicates
that CEM filters on two independent flow channels were capturing equivalent amounts of GOM.
This equivalency was true for both of the replicate systems (System A and B) that were each
simultaneously measuring flux from replicate trays of the same material. The high level of





replication displayed by the system, both in general and in detail, for multiple substrate types,
increases confidence that these measurements represent a real net surface exchange of GOM.
Positive GOM fluxes were associated with the higher substrate concentration materials (TCL,
TCT). The low Hg TCC experienced net GOM deposition, with a mean $V_d$ of $0.03 \pm 0.01$ cm s$^{-1}$
(n = 6). The 3x higher Hg concentration LTL showed either no net flux or slightly positive GOM
emission ($64 \pm 10$ pg m$^{-2}$ h$^{-1}$, n = 3). The very high substrate Hg concentration TCT$^-$ materials
showed uniformly high GOM emission ($4060 \pm 1000$ pg m$^{-2}$ h$^{-1}$, n = 7).
The mean ambient GOM in the greenhouse during this period was $130 \pm 55$ pg m$^{-3}$ (n = 24);
however, there was a distinct trend of increasing ambient GOM over the course of
measurements. During the TCC measurements, ambient GOM concentration in the greenhouse
was $70 \pm 20$ pg m$^{-3}$ (n = 8). This increased to $150 \pm 20$ pg m$^{-3}$ (n = 8) during the LTL
measurements, and up to $180 \pm 40$ pg m$^{-3}$ (n = 8) during the TCT measurement.

**291 3.3 GOM flux measurement over complete range of materials: Winter**

A full set of 24 h GOM flux measurements including all the available mining materials was
made in the following winter period (January – March, 2016). Ambient GOM concentration in
greenhouse air was 50 pg m$^{-3}$ (n = 16), much lower than in the summer months. Mean RH and
solar radiation were similar between the summer and winter periods, due to the attenuating effect
of the greenhouse. However, mean air and substrate temperatures were significantly lower in the
winter (Table 1).
All measured fluxes were above the $\Delta C_{GOM}$ detection limit (Fig. 7a). The relationship between
GOM flux measured on the A and B systems was slightly less than 1:1 (B = 0.87*A) and not as



strong ($r^2 = 0.74$) as during the summer measurements. Without chamber blank correction, GOM
fluxes were uniformly negative for all materials except the very high Hg TCT (Fig. 7b, orange
shades). However, with the chamber blank correction applied, GOM flux from the cap materials
became ambiguous (i.e. both deposition and emission observed) and positive for all leach and
tailings materials except the LTT (Fig. 7c, green shades).
Three replicated materials had consistent fluxes, with or without blank correction: GS cap (GOM
deposition, $V_d = 0.02 – 0.04$ cm s$^{-1}$, corrected), LTT (GOM deposition, $V_d = 0.01 – 0.05$ cm s$^{-1}$,
corrected), and TCT (strong GOM emission). The GS mine exploits a predominantly
carbonaceous ore deposit, and much of the mine and surrounding areas are subject to carbon
loading from aerial dust deposition. The carbon in the GS waste rock and ore likely facilitate
deposition of Hg and GOM (c.f. Miller et al, 2013 ; Eckley et al., 2011), . The LTT material is
from a non-active tailings impoundment that was partially revegetated at the time of collection,
versus the entirely barren surface typical of active tailings. Although the LTT Hg concentration
was high (11 µg g$^{-1}$), the surface was old, and it is possible this material behaved more as a
background substrate. The TCT material was collected from an actively filling tailings
impoundment that collected process waste from a variety of ore types, from multiple mine sites,
and had the highest Hg concentration of the materials used in this study, which likely explains its
tendency to by a strong emitter of both GEM and GOM, a conclusion also made by Eckley et al.
(2011a).
An interesting point is that background (2 m and inlet GOM concentrations were very similar
during the summer measurements (Fig. 8a), but inlet concentrations were 3x higher than
background during the winter measurements (Fig. 8b). The lower winter background GOM (50




pg m$^{-3}$) compared to summer (110 pg m$^{-3}$) may partially explain this, however the distance
between the inlet and background sampling heights was only ~ 1 m, which implies a strong
vertical gradient in GOM concentration in the greenhouse air. The explanation for this is less
well mixed air in the greenhouse bay during the winter, as the greenhouse circulation was shut
down at temperatures approaching 13 °C, setting up stratified conditions. The relatively high
inlet level GOM concentrations during the winter also caused the higher deposition observed in
the chamber blank measurements during this time.
**4   Conclusions**
This study presents direct GOM flux measurements using a CEM filter technique and provides
an analysis of whether the necessary measurements of small differences in GOM concentration
were possible using a DFC method.  Measurements of GOM flux were above calculated
detection limits in most cases, and both intra- (Tekran vs Pump sample) and inter- (A vs B)
system replicate measurements showed very good agreement. After initial trials at 72 h and 48 h,
a 24 h sample time was found to be generally sufficient for detecting GOM flux on the mining
materials used in this study, some of which were similar to background soils. It would be
possible to operate the system at higher flow rates, to decrease sample time and improve the
temporal resolution of the flux measurements. However, sample deployment and collection
require 20-30 min and to some extent perturbs the system, so ultimately the flux resolution is
limited by practical operational constraints. The 24 h measurement at least serves to capture net
flux over a full diel cycle without continuous interruptions.
We specifically refer to our measurement as GOM. This is because particulate Hg is not *emitted*
from a surface in the volatile sense, and particle entrainment was negligible at the low flow





velocities generated in the flux chambers. Thus GOM concentrations measured at the chamber
outlet CEM filters are dominated by GOM. If ambient air GOM concentrations were primarily
PBM, at a size fraction large enough to be captured by the CEM filters (i.e. > 0.8 μm), it will
undeniably be captured at the chamber inlet filters. However, PBM at this size would likely
deposit to the substrate surface within the chamber, thus deposition is GOM loading on the $C_o$
filters versus the $C_i$ filters.
Here we have demonstrated that GOM can be deposited and emitted from a surface. It is unlikely
GOM was produced by oxidation of GEM given short time of air moving through the chamber
and the fact that strong oxidants were likely removed as air moved through air handlers into the
greenhouse. This is a unique scientific finding.  Flux measured from low Hg, non-mineralized
cap materials for both GEM and GOM were low positive and negative values oscillating around
a net zero flux (Table 1). The observed GEM deposition velocities were typical of non-vegetated
surfaces, while GOM deposition occurred at $V_d$ values (0.01 – 0.07 cm s$^{-1}$) on the low end of the
suggested range (Zhang et al., 2009). The higher Hg concentration leach and tailings material
(except LT tails) showed net GEM and GOM emission in the summer, while deposition was
observed for both for most materials in the winter. The highest GOM emissions were
consistently observed for the highest Hg concentration substrate, TCT, in both summer and
winter conditions with fluxes higher in the summer during which time the greenhouse was better
mixed. The implications of these results are that GOM fluxes may be directly measured and that
contaminated areas such as mine tailings impoundments can act as a direct emission source of
GOM compounds to the atmosphere.

**Acknowledgements**



The authors would like to acknowledge funding from National Science Foundation Grant
629679, Barrick Gold Corp and Newmont Mining Corp for donating substrate materials, and Dr.
Ashley Pierce for invaluable assistance in preparing this manuscript.

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

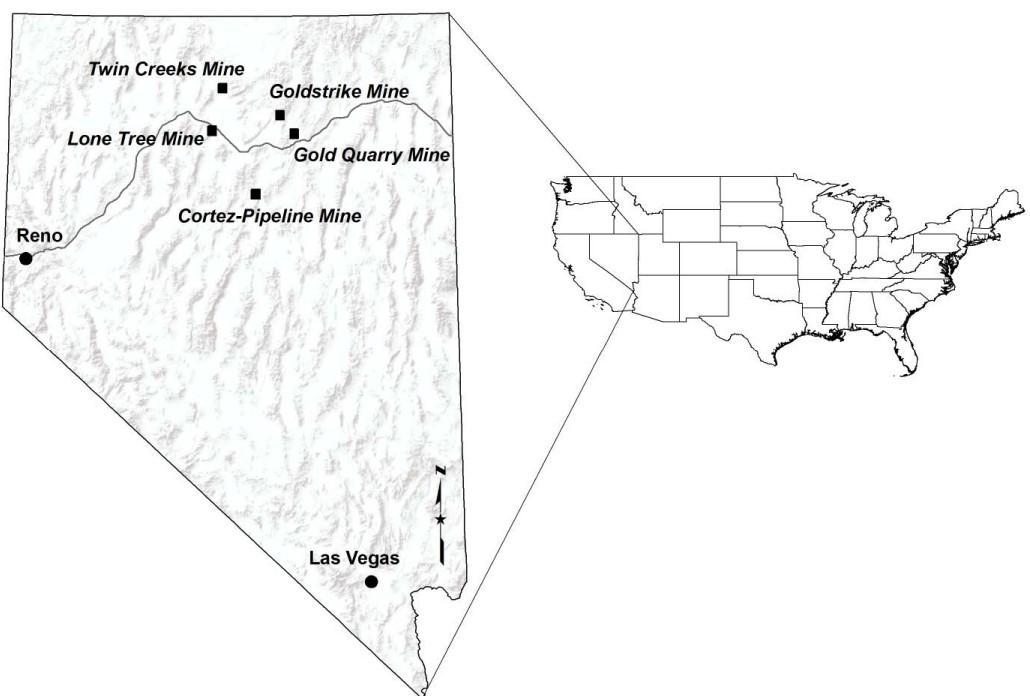

**Figure 1**.  Location map of study sites, Nevada, USA (From Miller and Gustin, 2013).





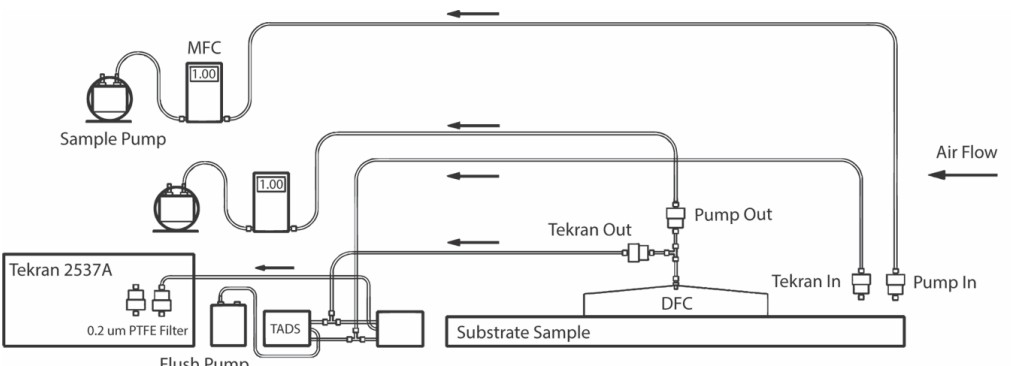


**Figure 2**. Diagram of one GOM filter-based flux system, deployed in duplicate as Systems A and B..Filter packs indicate the location of the CEM samples.


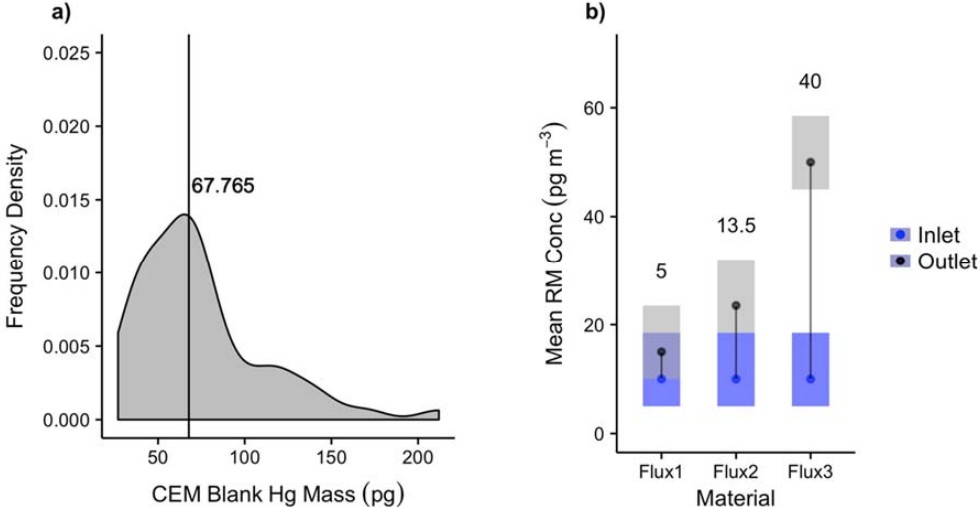


**Figure 3**. Determination of $\Delta C_{GOM}$ detection limit. a) Distribution of Hg mass on unused "blank" CEM filters (median = 68 pg) b) Hypothetical example of statistically detectable GOM flux criteria: shaded boxes represent the maximum uncertainty in concentration, based on 95% confidence interval around the median filter blank (58 – 73 pg), Flux1 represents an insufficiently resolvable $\Delta C_{GOM}$ in which the 95% confidence intervals around the median blank-corrected $C_o$ and $C_i$ values overlap, Flux2 represents the minimum detectable $\Delta C_{GOM}$ (13.5 pg m$^{-3}$), and Flux3 represents an obviously resolvable $\Delta C_{GOM}$.





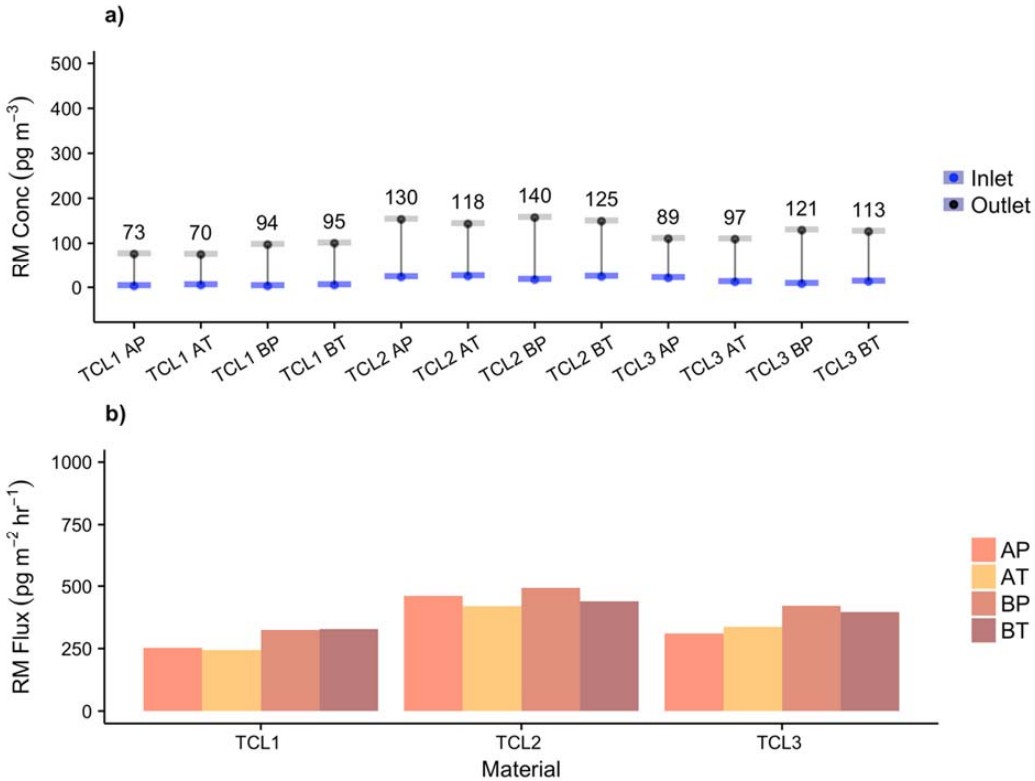


**Figure 4**. Test 72 h replicate measurements of TCL material. a) $\Delta C_{GOM}$ values: grey points indicate chamber
concentration ($C_o$), blue points indicate inlet air concentration ($C_i$), and the numeric value of $\Delta C_{GOM}$ is shown above
b) GOM flux from TCL material in three consecutive 72 h measurements, no chamber blank correction.  Sample
line labels: AP = Pump A, PB = Pump B, AT = Tekran A, BT = Tekran B.

544

545

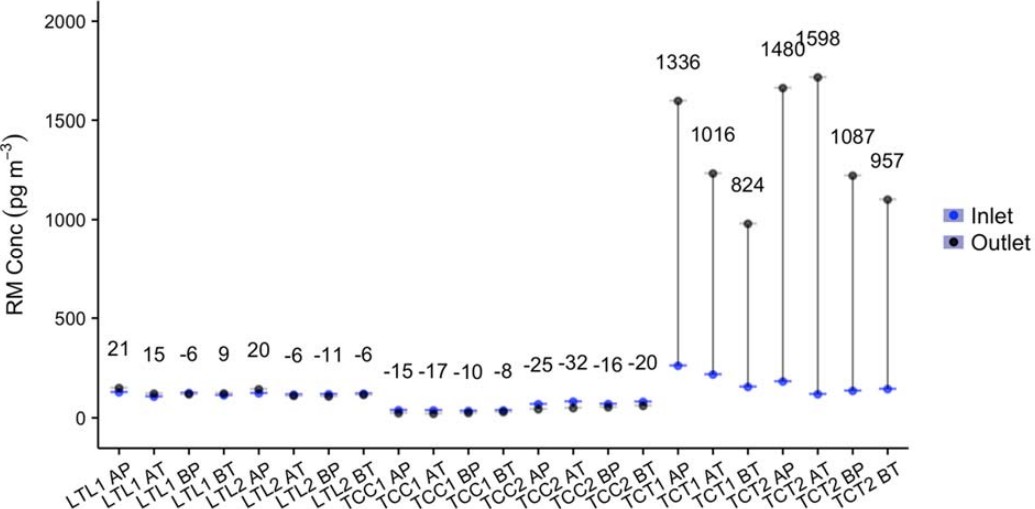

**Figure 5.** Expanded set of GOM flux measurements for select materials during summer. $\Delta C_{GOM}$ was below detection for 5 of 8 LTL measurements, and 2 of 8 TCC measurements, and these values were excluded.

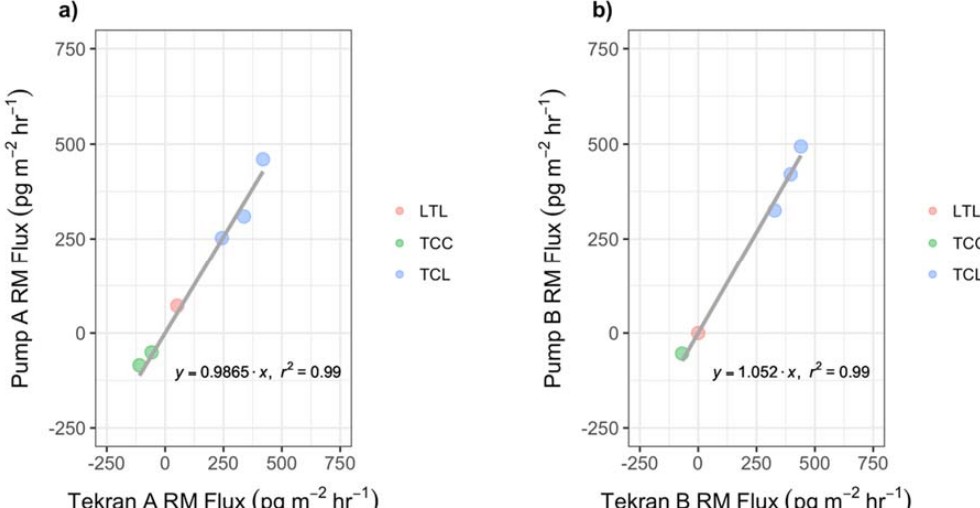

**Figure 6.** Comparison of GOM fluxes measured by Tekran controlled flow sample lines and external pump flow controlled sample lines, for a) System A and b) System B using TCL, TCC, LTL, and TCT summer measurements. Note TCT data not graphed, as fluxes were an order of magnitude higher and skew the regression $r^2$ towards 1.



**Figure 7**. GOM flux measurements for all materials, winter 2016. a) $\Delta C_{GOM}$, above detection limit for all measurements b) GOM flux, no chamber blank correction (shaded orange) c) GOM flux, with chamber blank correction (shaded green).




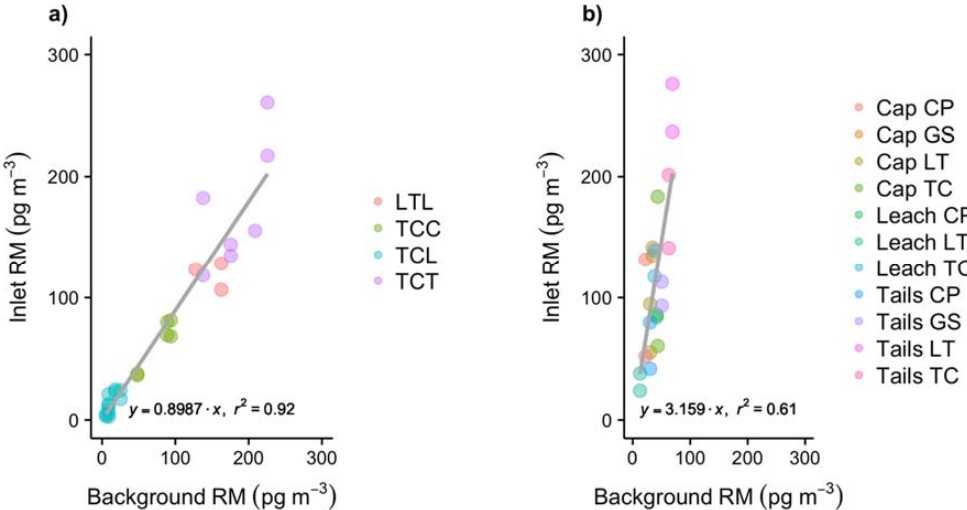

559

**Figure 8**. Comparison of ambient background GOM concentrations measured at 2 m height in the greenhouse, vs GOM concentrations measured at the chamber inlet, for a) Summer 2015, and b) Winter 2016. Background vs inlet concentrations were comparable during Summer measurements, but inlet concentrations were much higher relative to background in the winter.

564

565



**Table I.**

| | Material | Sample | Date | Substrate Conc (ng g⁻¹) | GEM Flux (ng m⁻² h⁻¹) | RM Flux (ng m⁻² h⁻¹) | RM Inlet (ng m⁻³) | RM $V_d$ (cm s⁻¹) | Ambient RM (ng m⁻³) | Temp (°C) | RH (%) | Solar (W m⁻²) | Soil Temp (°C) |
|---|---|---|---|---|---|---|---|---|---|---|---|---|---|
| | Leach | TCL1* | 5/16-5/19 | | 134 | 0.29 | 0.01 | - | 0.01 | | | | |
| | | TCL2* | 5/19-5/22 | 11900 | 151 | 0.45 | 0.03 | - | 0.03 | na | na | na | na |
| | | TCL3* | 5/22-5/25 | | 215 | 0.37 | 0.02 | - | 0.01 | | | | |
| Summer 2015 | Cap | TCC1A P | 7/21-7/23 | | 10 | -0.05 | 0.04 | 0.04 | | | | | |
| | | TCC1A T | | | | -0.06 | 0.04 | 0.04 | 0.05 | 22.6 | 41.6 | 15.8 | 23.9 |
| | | TCC1B P | | | 12 | na | 0.03 | na | | | | | |
| | | TCC1B T | | 230 | | na | 0.04 | na | | | | | |
| | | TCC2A P | 7/29-7/31 | | 4 | -0.09 | 0.07 | 0.03 | | | | | |
| | | TCC2A T | | | | -0.11 | 0.08 | 0.04 | 0.09 | 24.8 | 28.7 | 15.8 | 26.4 |
| | | TCC2B P | | | 13 | -0.05 | 0.07 | 0.02 | | | | | |
| | | TCC2B T | | | | -0.07 | 0.08 | 0.02 | | | | | |
| | Leach | LTL1A P | 8/12-8/14 | | 105 | 0.07 | 0.13 | - | | | | | |
| | | LTL1A T | | | | 0.05 | 0.11 | - | 0.16 | 23.8 | 30.2 | 17.4 | 26.5 |
| | | LTL1B P | | | 87 | na | 0.12 | na | | | | | |
| | | LTL1B T | | 590 | | na | 0.11 | na | | | | | |
| | | LTL2A P | 8/14-8/16 | | 100 | 0.07 | 0.12 | - | | | | | |
| | | LTL2A T | | | | na | 0.12 | na | 0.13 | 23.4 | 24.1 | 17.3 | 25.6 |
| | | LTL2B P | | | 99 | na | 0.12 | na | | | | | |
| | | LTL2B T | | | | na | 0.12 | na | | | | | |
| | Tailings | TCT1A P | 8/26-8/28 | | 483 | 4.46 | 0.26 | - | | | | | |
| | | TCT1A T | | | | 3.37 | 0.22 | - | 0.22 | 23.8 | 31.4 | 16.6 | 24.2 |
| | | TCT1B P | | | 370 | na | 0.15 | - | | | | | |
| | | TCT1B T | | 35750 | | 2.76 | 0.16 | - | | | | | |
| | | TCT2A P | 8/28-8/31 | | 702 | 5.02 | 0.18 | - | | | | | |
| | | TCT2A T | | | | 5.43 | 0.12 | - | 0.16 | 22.0 | 33.3 | 13.9 | 22.1 |
| | | TCT2B P | | | 457 | 3.02 | 0.13 | - | | | | | |
| | | TCT2B T | | | | 2.57 | 0.14 | - | | | | | |
| Winter 2016 | Cap | TCCA | 1/6/16 | 230 | 1 | -0.24 | 0.18 | 0.07 | 0.04 | 15.0 | 37.6 | 2.7 | 14.5 |
| | | TCCB | | | 1 | 0.05 | 0.06 | - | | | | | |
| | | LTCA | 1/7/16 | 150 | 0 | 0.00 | 0.09 | - | 0.03 | 14.5 | 38.6 | 9.7 | 11.8 |
| | | LTCB | | | -1 | 0.07 | 0.06 | - | | | | | |
| | | CPCA | 1/20/16 | 120 | -1 | 0.00 | 0.13 | - | 0.02 | 14.5 | 31.7 | 12.9 | 15.6 |
| | | CPCB | | | na | 0.06 | 0.05 | - | | | | | |
| | | GSCA | 1/21/16 | 200 | -1 | -0.12 | 0.13 | 0.02 | 0.03 | 14.6 | 26.6 | 11.0 | 15.8 |
| | | GSCB | | | -1 | -0.20 | 0.14 | 0.04 | | | | | |
| | Leach | TCLA | 1/26/16 | 11900 | 29 | 0.11 | 0.14 | - | 0.04 | 13.4 | 35.3 | 9.3 | 15.2 |
| | | TCLB | | | 30 | 0.11 | 0.12 | - | | | | | |
| | | CPLA | 1/29/16 | 310 | -18 | 0.00 | 0.08 | - | 0.04 | 13.9 | 34.9 | 16.5 | 15.3 |
| | | CPLB | | | -12 | 0.00 | 0.09 | - | | | | | |
| | | LTLA | 2/2/16 | 590 | -4 | 0.13 | 0.02 | - | 0.01 | 11.7 | 25.5 | 14.6 | 14.2 |
| | | LTLB | | | -3 | 0.07 | 0.04 | - | | | | | |
| | Tailings | CPTA | 2/9/16 | 21150 | 4 | 0.00 | 0.08 | - | 0.03 | 14.1 | 37.5 | 18.9 | 15.7 |
| | | CPTB | | | 5 | 0.25 | 0.04 | - | | | | | |
| | | GSTA | 2/10/16 | 6960 | 16 | 0.16 | 0.11 | - | 0.05 | 14.4 | 39.3 | 11.6 | 13.8 |
| | | GSTB | | | -1 | 0.11 | 0.09 | - | | | | | |
| | | LTTA | 3/20/16 | 11020 | 2 | -0.45 | 0.28 | 0.05 | 0.07 | 19.8 | 23.1 | 31.0 | 22.3 |
| | | LTTB | | | 2 | -0.08 | 0.24 | 0.01 | | | | | |
| | | TCTA | 3/22/16 | 35750 | 51 | 0.52 | 0.20 | - | 0.06 | 17.0 | 26.6 | 37.4 | 19.3 |
| | | TCTB | | | 71 | 0.79 | 0.14 | - | | | | | |

**Table I.** Summary of GEM and GOM flux and ambient parameters for all measurements. Fluxes are chamber blank-corrected where applicable, and $V_d$ values are based on corrected fluxes (- indicates no deposition, *na* indicates non-detectable flux).