# Peer review of "Reactive mercury flux measurements using cation exchange membranes"

_Atmospheric Measurement Techniques, 2018_

## Referee Comment (RC1) · Anonymous Referee #1 · 11 Jan 2019

This work addresses a significant research gap and provides a new, and seemingly improved methodology for measuring gaseous oxidized Hg fluxes from source material to air. This work has wide-ranging applicability in estimating Hg fluxes from contaminated materials, surface waters, Arctic permafrost, etc. Because it has such a wide applicability, it would be beneficial to researchers if the authors included in the discussion how their findings of both the method's abilities and limitations apply to other environmental settings. For instance, providing information on estimated sample time based on Hg substrate concentration, and detection limits in context of background Hg concentrations would be extremely useful. Overall the manuscript is well written, however it would greatly benefit from a streamlining of the discussion around methodology. Most of the information provided in Sections 2.4 and 2.5 is interesting but distracts

from the scientific findings and would be better suited as supplementary material or moved to other parts of the discussion. Finally, it can be hard to keep track of all the abbreviations/acronyms used in this study, a legend of sorts would be helpful.

Specific comments:

L54: add a couple sentences regarding the reactivity of GOM and how this relates to its residence time in the air compared to GEM, and how this relates to the need for accurate active sampling of GOM in a variety of environments

L84: it would be beneficial here to briefly discuss limitation to passive CEM sampling and what gap in understanding measuring fluxes could fill – make it clear to the reader how this method could be applied

L99: include Hg content of source material

LL114-L116: discussion of chamber footprint is confusing here, suggest moving this sentence to where you discuss need to place the flux chamber directly on the surface conditions ($\sim$L240)

L163: How did you determine how often filters would be changed? This is later mentioned in the discussion, however it is unclear whether sampling time was determined based on previous measurements or purely based on substrate Hg concentrations.

L201: define 'high' ambient GOM concentrations

LL208-214: "All fluxes were measured in . . .." This section should be with the experimental set-up.

L260: what's accounting for the 21.4% variation in mean GOM flux measured from the TCL material? Could changing conditions in greenhouse explain any of this variability?

L265: what's considered 'good agreement' - how does this compare to other methods (i.e. KCl denuders or other membranes)

LL281-285: A summer flux figure (similar to Figure 7b,c) would be helpful for comparing summer to winter values

L325: this would be a good place to discuss how CEM GOM measurements are affected by temperature and humidity, and what accompanying physiochemical and meteorological measurements should be taken with GOM and GEM measurements

L331: define small difference

LL342-362: this seems out of place here, and should be moved to methodology section where defining GOM and discussing sample inlet and filter size (L163)

Technical corrections:

L214: (see below) - Is this referring to a figure or text?

L290: include reference to Table 1

L300: should this reference Figure 8?

L319: word choice, suggest using 'observation' instead of 'point'

Figure 2: it is not clear where the filter packs are located in the schematic. Also remove double period in figure caption '..'

Table 1: is substrate concentration total Hg conc.? If so, use the same units as reported in the text ($\mu$g g-1) and label as Hg. If not Hg, then please include Hg conc.

---

## Short Comment (SC1) · 15 Jan 2019

This is a well-written study in mercury chemistry and geochemical cycling across the atmosphere-terrestrial boundary, an area of much uncertainty and debate. Nonetheless, Flux measurements are difficult measurements to make and often produce variable results in regularly monitored gases (e.g. $CO_2$) due to variability in the measurements of micro- and macro- movement of air (vertically and horizontally) and the gases themselves (1,2,3). The measurement concerns of these parameters stem from differing instruments (and their quality assurance and control protocols), data handling, corrections, and calculations, and even differing personnel (1,2,3). Considering the current study focusses the ultra-trace level fluxes of GOM (ppqv) that the authors themselves discuss as being notoriously difficult to measure just in terms of concentrations,

[Figure]

I have concerns about the validity of the results and the conclusion made. As such, for the following reasons I would have to reject the study:

(i). What is the mechanism for GOM emissions from the substrate? There is no discussion on what might be driving low-volatility GOM compounds from a relatively stable state in the solid phase, sorbed (physically or chemically) to the substrate into the gaseous phase as Hg2+. This is difficult to envisage thermodynamically. There is substantial discussion in the literature of Hg2+ photoreduction or biotic reduction and re-emission from soils and aqueous bodies, but the flux of Hg is Hg0 not Hg2+ (e.g. 4,5,6). I cannot find literature describing the scenario required for the authors' conclusions, which leads to point (ii).

(ii). In lines 161-163 the authors state that particle entrainment is not expected. While the ideal modeled scenario shows reasonable laminar flow within the flux chamber itself (7), perfectly laminar, non-turbulent flow is less likely real deployments due to substrate, chamber, and flow imperfections. Furthermore, the ideal modeled scenario in Eckley et al. (7) shows the highest flow rates within the chamber are always at the surface of the substrate. These factors suggest there is potential for the chamber to generate suspension of particles. These particles (potentially carrying mercury) can then stick to the CEMs. This may include particles finer than 0.8 $\mu$m if they display any affinity for the CEM. If this occurs then the CEMs are not measuring GOM fluxes, but are instead measuring a GOM/PBM signal generated by the chamber itself. Only a small number of particles from these heavily contaminated substrates will cause a significant artefact in the flux signal. This must be categorically confirmed or denied before these fluxes can be discussed further. High resolution microscopy to observe the surface of the CEMs before and after deployment may present a means to determine the presence of any particles.

(iii). The uncertainty of the sampler is not rigorously described for GOM air concentrations, let alone for flux measurements. Uncertainty in this study seems to be based solely on the median and confidence intervals of the CEM filter blanks. This in itself is a

problem. Figure 3 shows, as stated by the authors, the blank data to be heavily skewed to the right (as would be expected for blanks). There are blank CEMs with over 200 pg, which would cause a flux measurement with a CEM carrying this much residual Hg to be overstated. The median results in a lower value than the mean in determining the central tendency of the blanks. The mean would be more appropriate to capture the elevated blank concentrations that do exist on some CEMs. All the data on the mass of Hg in blanks and the samples (not just fluxes, but include pg of Hg per CEM) should be included in the supplemental so readers can make a better assessment of the data as a whole. Moving on, what is the overall uncertainty of the CEMs for measuring GOM? What is the accuracy and precision of these measurements? Without this overall uncertainty (and solely the blank uncertainty) we cannot be sure the differences do not fall within the uncertainty of the full sampling methodology.

(iv). I do not agree with the removal of 44% (7/16) of measurements for LTL and TCC sediments because the results were below detection limits. These remain results and should be included in the analysis and discussion.

(v). GOM or RM? There are references to both these descriptors within the manuscript. All the figure captions and axes labels refer to RM, yet throughout the text the authors make the case for GOM. Maybe this is an oversight, but quite an important one considering the context of the study and does suggest the authors have had some back-and-forth on which term is most appropriate.

Literature:

1. Foken, T., & Wichura, B. (1996). Tools for quality assessment of surface-based flux measurements. Agricultural and forest meteorology, 78(1-2), 83-105.

2. McGillis, W. R., Edson, J. B., Ware, J. D., Dacey, J. W., Hare, J. E., Fairall, C. W., & Wanninkhof, R. (2001). Carbon dioxide flux techniques performed during GasEx-98. Marine Chemistry, 75(4), 267-280.

3. Massman, W. J., & Lee, X. (2002). Eddy covariance flux corrections and uncertainties in long-term studies of carbon and energy exchanges. Agricultural and Forest Meteorology, 113(1), 121-144.

4. Lindberg, S. E., Kim, K. H., Meyers, T. P., & Owens, J. G. (1995). Micrometeorological gradient approach for quantifying air/surface exchange of mercury vapor: tests over contaminated soils. Environmental science & technology, 29(1), 126-135.

5. Fritsche, J., Obrist, D., & Alewell, C. (2008). Evidence of microbial control of Hg0emissions from uncontaminated terrestrial soils. Journal of plant nutrition and soil science, 171(2), 200-209.

6. Yin, R., Feng, X., Chen, B., Zhang, J., Wang, W., & Li, X. (2015). Identifying the sources and processes of mercury in subtropical estuarine and ocean sediments using Hg isotopic composition. Environmental science & technology, 49(3), 1347-1355.

7. Eckley, C. S., Gustin, M., Lin, C. J., Li, X., & Miller, M. B. (2010). The influence of dynamic chamber design and operating parameters on calculated surface-to-air mercury fluxes. Atmospheric Environment, 44(2), 194-203.

---

## Short Comment (SC2) · 15 Jan 2019

CLARIFICATION: In my previous comment there was a mistake in point (ii). The highest flow rates were not observed at the surface of the substrate in Eckley et al. (2010). But the point remains ideal flow conditions are unlikely in real deployments and there is potential for some particle entrainment. The concern of artifacts from sorbed particles requires resolution by the suggested mechanism or otherwise; at the elevated Hg concentration of the substrate and ultra-trace GOM fluxes, any uptake of particulates to the CEMs would cause greatly effect the GOM flux data.

---

## Referee Comment (RC2) · Anonymous Referee #2 · 10 Feb 2019

General Comments

This manuscript details mercury flux measurements made over various mining waste materials using dynamic flux chambers (DFC). The main motivation for the paper involves the use of polysulfone cation exchange membranes on the DFC sample lines as a means of collecting gaseous oxidized mercury (GOM) and thereby making estimates of GOM flux from and to the mining waste materials.

After careful reading and consideration of the manuscript, I am afraid that I find the manuscript flawed and that the main conclusions of the paper are built around assumptions of potentially interfering processes that were unmeasured and/or are not as easily dismissed in reality as assumed by the authors.

Fundamentally, it is unclear to me how GOM can be emitted from these solid surface materials when nearly all studies find that GOM fluxes are dominantly in the deposition direction. The authors also do not explore in their paper a possible mechanism by which GOM emission could occur. The literature is rich in examples of how GOM is easily and quickly removed from the atmosphere to surfaces (deposited) because of its high reactivity and strong binding with surfaces. The authors point to previous work on GOM flux measurement in their introduction (lines 69-75), but even though a very small number of these studies (e.g., Skov et al. 2006) potentially found small GOM emissions, authors of previous works have attributed those emissions to measurement artifacts and/or the quick oxidation of gaseous elemental mercury emissions, making it merely appear that the fluxes are GOM. The Engle et al. (2005) paper does suggest that GOM emissions may occur with some sort of heterogeneous surface reaction. The authors state their findings are "a unique scientific finding" (line 363), but I feel there are too many confounding factors to confidently believe that this is likely the case. The specifics of why I would say this include: 1. As a first approximation to the suitability of this method for measurement of GOM fluxes, one would expect this method to be assessed against another relatively accepted method used in the published literature to assess fluxes. I could see this as unnecessary if all potentially uncontrolled variables were accounted for, but this is not the case in this paper. 2. The authors rely on assumptions based on previous modelling of the DFC that flow rates are too small to induce the entrainment and potential capture of particles. This seems an overly sweeping comment to me, especially since 1) smaller particles are more easily entrained than larger ones and thus the air flow rate to entrain particles would be particle size-specific and 2) there is no microscopic evidence given at all that particles were not trapped. To me, particle entrainment and capture is actually the most straightforward and most likely reason for what appeared to be, but quite possibly was not, GOM emission. In the paper, the authors have also labeled all of the GOM flux axes in their figures as "reactive mercury" (RM) fluxes, though they refer to the same fluxes in the text as GOM. As per the authors' own definition, RM includes both GOM and particulate bound mercury

(PBM). This is again suggestive that particulate mercury fluxes may be what are pushing the seeming GOM fluxes. 3. The authors do not suggest a mechanism by which GOM emission could occur. 4. The authors' handling of detection limits is somewhat puzzling and thus it is difficult to be confident that what is portrayed as a detectable flux, really is above the detection limit. Specifically, the authors use the median mass of Hg on CEM filters to arrive at a "practical detection limit". Normally, one would use a measure of variability to arrive at a detection limit, not just a central tendency. From Figure 3, it is apparent that the blank variability is fairly high. If the authors used 3X (limit of detection) and 10X (quantification limit) standard deviation of blank measurements, I would be interested to know how the ability to detect differences in inlet and outlet concentrations would be affected.

Specific Comments

Is it the norm for this journal to not have a "Discussion" section? I found it odd for the Results section to be followed by a Conclusions section.

Line 52-53: "Deposition rates" for GEM are indeed quite variable and dependent on a number of things. This is sentence is oddly written. A low deposition rate does already infer a long residence time and vice versa. Why give a range in "atmospheric lifetime" (including 1 minute), but say that deposition is slow?

Line 55: Does just "rapid deposition of GEM" "suggests that deposition and re-emission on short time scales is an important process driving movement."?

Line 69-73: This is a very large number of articles to reference for a single point. Moreover, many of these articles do not actually directly measure GOM fluxes, but rather sometimes just deposition via passive surfaces and sometimes just concentrations.

Lines 77-79: Is the overall suggestion here that GOM fluxes in a specific direction are important outside of AMDE's but that we just don't know much about it? I'm not a big fan of vague ending sentences like this unless you are indeed going to improve our

"understanding of GOM air-surface exchange PROCESSES".

Line 85: What is the relevance of the unpublished work referenced here?

Lines 97-100: While I did not find the abbreviations overly used in the paper, I don't really see their use. Are the different places on lines 97-98 even important? They are not really differentiated by "place" in the paper. The different types of material are important, but for example, is it really all that difficult to write "cap material" or "heap leach material" or certainly "tailings"? I could see the usefulness of abbreviations in some of the figures, but that important abbreviations could simply be defined in the captions.

Line 114: The authors refer to "previous watering experiments". They should verify that these experiments do not (hopefully) impart any variability into their experimental design.

Line 178: Do you really mean REACTIVE Hg flux? Term mismatch?

Line 204: What mechanistically would cause this to happen in the winter but not the summer? This nuanced way of dealing with blanks does not provide much assurance about control.

Line 234: This is a fairly bold statement, without an explanation as to why it "moves us a step further for understanding GOM flux."

Line 279: Not sure what you mean here by "both in general and in detail".

Line 294: Was it really "much" lower? It would appear statistically insignificantly different from the period at line 289, for TCC measurements, which says it's about 70 +/- 20 pg/m3.

Line 317: After looking at this Eckley et al. (2011a) paper, it appears that they measured only total gaseous mercury fluxes and not anything to delineate GEM from GOM. I also do not see anything about GOM at all in the conclusions of this paper. Your statement here therefore appears incorrect.

Line 335: Is this part about "generally sufficient" really a viable conclusion? Many of the 24-hour measurements on the lower concentration materials were not reconcilable (e.g., figure 5). There are also concerns about the handling of detection limits as per my previous comments.

Line 337: Until an absence of particles can be confirmed, I would not advocate for increasing the flow rate.

Line 343-344: That particles were not entrained is more an assumption than a conclusion in this paper. There were no direct measurements.

Line 344-345: GOM concentrations are dominated by GOM?

Line 346: The authors seem to be conflating PBM as a possible fraction of GOM, which it is not.

Line 347: I'm not sure I follow the writing in the paragraph exactly, because what about small particles (i.e. < 0.8 microns)? Is this paragraph about trying to extract out the possibility of a particulate Hg influence? It does not succeed as currently written.

Line 350: Just "unlikely"? This is a conclusions paragraph and none of these measurements were made.

---

## Author Comment (AC1) · 18 Feb 2019

Response to Anonymous Referee #1 This work addresses a significant research gap and provides a new, and seemingly improved methodology for measuring gaseous oxidized Hg fluxes from source material to air. This work has wide-ranging applicability in estimating Hg fluxes from contaminated materials, surface waters, Arctic permafrost, etc. Because it has such a wide applicability, it would be beneficial to researchers if the authors included in the discussion how their findings of both the method's abilities and limitations apply to other environmental settings. For instance, providing information on estimated sample time based on Hg substrate concentration, and detection limits in context of background Hg concentrations would be extremely useful.

Response: We thank this reviewer for the acknowledgement of the fact this is a new method that holds significant promise. We have added the information the reviewer requested at the end of the discussion section. We also thank this reviewer for their careful review and suggestions for improvement.

Overall the manuscript is well written, however it would greatly benefit from a streamlining of the discussion around methodology. Most of the information provided in Sections 2.4 and 2.5 is interesting but distracts from the scientific findings and would be better suited as supplementary material or moved to other parts of the discussion.

Response: We prefer to leave these two sections in the paper for they are important information regarding the method development.

Finally, it can be hard to keep track of all the abbreviations/acronyms used in this study, a legend of sorts would be helpful.

Response: We have eliminated some of the abbreviations that were not necessary. We also wrote out the names of the materials used. We included definition of the abbreviations in between the abstract and the introduction.

Specific comments: L54: add a couple sentences regarding the reactivity of GOM and how this relates to its residence time in the air compared to GEM, and how this relates to the need for accurate active sampling of GOM in a variety of environments

Response: Thank you very much for this comment. This information has been added and the paragraph rewritten so it flows better.

L84: it would be beneficial here to briefly discuss limitation to passive CEM sampling and what gap in understanding measuring fluxes could fill – make it clear to the reader how this method could be applied

Response: We prefer to not to discuss passive sampling here since this paper is focused on active sampling and I think this would be confusing to the readers.

L99: include Hg content of source material

Response: This information is in Table 1. We made this clearer. LL114-L116: discussion of chamber footprint is confusing here, suggest moving this sentence to where you discuss need to place the flux chamber directly on the surface conditions (_L240)

Response: We prefer to leave this sentence where it is. It seems to fit better where it is.

L163: How did you determine how often filters would be changed? This is later mentioned in the discussion, however it is unclear whether sampling time was determined based on previous measurements or purely based on substrate Hg concentrations.

Response: Flow rates were described in the paper for each type of test and varied from 72 to 24 hours. For materials where fluxes were not above the detection limit given the sampling time it is clear a longer sampling time was needed.

L201: define 'high' ambient GOM concentrations

Response: We have removed this sentence. It did not make sense.

LL208-214: "All fluxes were measured in : : :." This section should be with the experimental set-up.

Response: We have moved this as the reviewer requested.

L260: what's accounting for the 21.4% variation in mean GOM flux measured from the TCL material? Could changing conditions in greenhouse explain any of this variability?

Response: The 21.4% variability between tubs of material is due to the fact that are slight variations in concentrations of the overall material being measured. The good replication for flux measurements on one material demonstrates that the individual tub measurements were consistent. We have added this to the manuscript.

L265: what's considered 'good agreement' - how does this compare to other methods

(i.e. KCl denuders or other membranes)

Response: We removed good agreement for this component of the sentence was not necessary. r-friendly version Discussion paper LL281-285: A summer flux figure (similar to Figure 7b,c) would be helpful for comparing summer to winter values

Response: These are compared but not in as much detail in Figure 8. We prefer not to add an additional figure. We already have many. If the Editor wants us to do this, then we will.

L325: this would be a good place to discuss how CEM GOM measurements are affected by temperature and humidity, and what accompanying physiochemical and meteorological measurements should be taken with GOM and GEM measurements

Response: We thank the reviewer for this comment and we have added the following to the paper. Limited work in laboratory experiments with HgBr2 demonstrated that RH did not affect measurement of GOM by the CEM (Miller et al., 2019). No tests have been done to directly address if there is a temperature effect. When making these measurements scientists should measure RH and temperature inside and outside the chamber. Other information that would be useful include soil moisture, precipitation, and solar radiation.

L331: define small difference

Response we have added $\sim$10 pg and referred the reader to Figure 7a.

LL342-362: this seems out of place here, and should be moved to methodology section where defining GOM and discussing sample inlet and filter size (L163)

Response: We have changed the wording of the first sentence and prefer to keep this information here to reinforce our conclusion.

Technical corrections: L214: (see below) - Is this referring to a figure or text?

Response: To address this we changed this to read see Development of Method description below.

L290: include reference to Table 1

Response: Thanks for finding this we added reference to Table 1.

L300: should this reference Figure 8?

Response: No the data are in Figure 7, but the regression is not shown. We refer to Figure 7 as the location of the data used to derive the regression equation.

L319: word choice, suggest using 'observation' instead of 'point'

Response: We made this change.

Figure 2: it is not clear where the filter packs are located in the schematic. Also remove double period in figure caption '..'

Response: We think it is clear where the filter packs are they have a very distinct shape. Got rid of the extra period.

Table 1: is substrate concentration total Hg conc.? If so, use the same units as reported in the text (_g g-1) and label as Hg. If not Hg, then please include Hg conc.

Response: Yes, substrate concentration is total Hg concentration. We have changed the units in the text to reflect those in the table ng g-1 and inserted in the figure caption that Substrate conc. is total Hg concentration.

---

## Author Comment (AC2) · 18 Feb 2019

This is a well-written study in mercury chemistry and geochemical cycling across the atmosphere-terrestrial boundary, an area of much uncertainty and debate.

Response: Thank you.

Nonetheless, Flux measurements are difficult measurements to make and often produce variable results in regularly monitored gases (e.g. $CO_2$) due to variability in the measurements of micro- and macro- movement of air (vertically and horizontally) and the gases themselves (1,2,3). The measurement concerns of these parameters stem from differing instruments (and their quality assurance and control protocols), data handling, corrections, and calculations, and even differing personnel (1,2,3). Considering the current study focusses the ultra-trace level fluxes of GOM (ppqv) that the authors themselves discuss as being notoriously difficult to measure just in terms of concentrations, I have concerns about the validity of the results and the conclusion made.

Response: We would like to point out here that all experiments were done by one operator Matthieu Miller who has made many flux measurements from soils over many years and the same sampling procedure was carefully applied for each measurement.

As such, for the following reasons I would have to reject the study:

(i). What is the mechanism for GOM emissions from the substrate? There is no discussion on what might be driving low-volatility GOM compounds from a relatively stable state in the solid phase, sorbed (physically or chemically) to the substrate into the gaseous phase as Hg2+. This is difficult to envisage thermodynamically. There is substantial discussion in the literature of Hg2+ photoreduction or biotic reduction and re-emission from soils and aqueous bodies, but the flux of Hg is Hg0 not Hg2+ (e.g. 4,5,6). I cannot find literature describing the scenario required for the authors' conclusions, which leads to point (ii).

Response: The mechanism is volatilization. Volatilization of GOM compounds from pure salts has been demonstrated in many studies (see a few refs below). In fact, we can load membranes with salts of GOM compounds by placing them above a container holding them and then verify the compound emitted using an ion chromatograph. We have added some references that point to this in the paper in the introduction.

Finley, B. D.; Jaffe, D. A.; Call, K.; Lyman, S. N.; Gustin, M., Development, testing, and deployment of an air sampling manifold for spiking elemental and oxidized mercury during RAMIX. Submitted to Environ. Sci. Technol. 2013.

Landis, M. S.; Stevens, R. K.; Schaedlich, F.; Prestbo, E. M., Development and Characterization of an Annular Denuder Methodology for the Measurement of Divalent Inorganic Reactive Gaseous Mercury in Ambient Air. Environ. Sci. Technol. 2002, 36, (13), 3000-3009.

Lyman, Seth; Jones, Colleen; O'Neil, Trevor; Allen, Tanner; Miller, Matthieu; Gustin, Mae; Pierce, Ashley; Luke, Winston; Ren, Xinrong; Kelley, Paul (2016) Automated Calibration of Atmospheric Oxidized Mercury Measurements Environmental Science and Technology 50 12921-12927 DOI: 10.1021/acs.est.6b04211 Huang J., Miller M.B., Weiss-Penzias P., Gustin M.S. 2013 Comparison of Reactive Mercury Measurements Made with KCl-coated Denuders, Nylon Membranes, and Cation Exchange Membranes Environmental Science and Technology, 47: 7307-7316. DOI: 10.1021/acs.est.5b00098 McClure, C. D., Jaffe, D. A., Edgerton, E.S.: Evaluation of the KCl Denuder Method for Gaseous Oxidized Mercury using HgBr2 at an In-Service AM-Net Site, Environ. Sci. Technol., 48 (19), 11437-11444, 2014. Here is the text added: "The potential for GOM volatilization from surfaces has not been quantified; however, we know that GOM can be emitted from salts of a variety of GOM compounds including HgCl2, HgBr2, HgO, HgSO4, and Hg(NO3)2 (Finley et al., 2013; Landis et al., 2002; Lyman et al., 2016; Huang et al., 2013; McClure et al., 2014). Because of rapid reactions observed during the Reno Atmospheric Mercury Intercomparison Experiment (RAMIX) (Gustin et al., 2013), methods to measure GOM at a short time resolution are needed."

(ii). In lines 161-163 the authors state that particle entrainment is not expected. While the ideal modeled scenario shows reasonable laminar flow within the flux chamber itself (7), perfectly laminar, non-turbulent flow is less likely real deployments due to substrate, chamber, and flow imperfections. Furthermore, the ideal modeled scenario in Eckley et al. (7) shows the highest flow rates within the chamber are always at the surface of the substrate. These factors suggest there is potential for the chamber to generate suspension of particles. These particles (potentially carrying mercury) can then stick to the CEMs. This may include particles finer than 0.8 _m if they display any affinity for the CEM. If this occurs then the CEMs are not measuring GOM fluxes, but are instead

measuring a GOM/PBM signal generated by the chamber itself. Only a small number of particles from these heavily contaminated substrates will cause a significant artefact in the flux signal. This must be categorically confirmed or denied before these fluxes can be discussed further. High resolution microscopy to observe the surface of the CEMs before and after deployment may present a means to determine the presence of any particles.

Response: An adaptation of the below has been added to the paper.

Based on the computational fluid dynamic modeling in Eckley at al. (2010), at 1 Lpm there is turbulence in the DFC, mainly near the entrances and exit. Based on the dimensions of the DFC, the friction velocity was on the order of 10-4 to 10-3 m s-1 in the flow range of 1 to 2 Lpm (Professor Jerry Lin, Lamar University-College of Engineering, Personal communication 18 February 2019). Gillette (1988) reported, in a paper that focused on trying to understand the generation of dust, threshold friction velocities for agricultural soils. Threshold friction velocity for wind erosion corresponds to the minimum wind stress needed to overcome forces holding soil particles in place. Experiments in Gillette (1988) were done using a portable wind tunnel using a variety of soils. (Dr. Heather Holmes, University of Nevada-Physics, Personal communication, 18 February 2019).

Based on this information, friction velocity for the flux chamber under operating conditions during our experiment were 2 orders of magnitude lower than those determined for a variety of soils (Gillette, 1988). In addition, as mentioned in the paper, at the onset of the GOM flux experiments, substrates were undisturbed for ∼3 years, and were completely dry and well compacted/consolidated from previous watering experiments. Lastly, similar values were obtained for repeated experiments. Based on these pieces of information the potential for particle entrainment and suspension in the chamber was unlikely, and it is thought volatilization of GOM was occurring. Future work should work on further investigating this. Gillette, D.A., 1988. Threshold friction velocities for dust production for agricultural soils. Journal of Geophysical Research 93, 12645–12662.
We do not need to respond to the rest of the comment, because the reviewer corrected his comment in a second response. McLagan second comment "CLARIFICATION: In my previous comment there was a mistake in point (ii). The highest flow rates were not observed at the surface of the substrate in Eckley et al. (2010). But the point remains ideal flow conditions are unlikely in real deployments and there is potential for some particle entrainment. The concern of artifacts from sorbed particles requires resolution by the suggested mechanism or otherwise; at the elevated Hg concentration of the substrate and ultra-trace GOM fluxes, any uptake of particulates to the CEMs would cause greatly effect the GOM flux data."

(iii). The uncertainty of the sampler is not rigorously described for GOM air concentrations, let alone for flux measurements. Uncertainty in this study seems to be based solely on the median and confidence intervals of the CEM filter blanks. This in itself is a problem. Figure 3 shows, as stated by the authors, the blank data to be heavily skewed to the right (as would be expected for blanks). There are blank CEMs with over 200 pg, which would cause a flux measurement with a CEM carrying this much residual Hg to be overstated. The median results in a lower value than the mean in determining the central tendency of the blanks. The mean would be more appropriate to capture the elevated blank concentrations that do exist on some CEMs. All the data on the mass of Hg in blanks and the samples (not just fluxes, but include pg of Hg per CEM) should be included in the supplemental so readers can make a better assessment of the data as a whole. Moving on, what is the overall uncertainty of the CEMs for measuring GOM? What is the accuracy and precision of these measurements? Without this overall uncertainty (and solely the blank uncertainty) we cannot be sure the differences do not fall within the uncertainty of the full sampling methodology.

Response: We feel the median is more appropriate since the mean captures outliers and these were not very abundant as demonstrated in Figure 3. We have added the following regarding the accuracy and precision. GOM was determined after digestion in an oxidizing acid solution, reduction to Hg0, gold amalgamation, and final quantification

by cold vapor atomic fluorescence spectrometry (CVAFS, EPA Method 1631, Rev. E) using a Tekran® 2600 system. The system background Hg signal was determined for every analytical run by analyzing pure reagent solution in the same vials and at the same volume as used for actual filter samples. Total Hg standards (5 to 100 ppb) were analyzed before and after each batch of 10 filter samples to check precision and recovery, and the mean recovery for all Hg standards was 97.2 $\pm$ 5.0 %. Coefficient of variation in concentrations for triplicate filters is typically 1 to 10% with the low value being associated with concentrations > 100 pg.

(iv). I do not agree with the removal of 44% (7/16) of measurements for LTL and TCC sediments because the results were below detection limits. These remain results and should be included in the analysis and discussion.

Response: Since we were not focusing on scaling up fluxes, but instead were just attempting to understand GOM fluxes we felt removal of data below the detection limit was appropriate. We added a sentence to that affect.

(v). GOM or RM? There are references to both these descriptors within the manuscript. All the figure captions and axes labels refer to RM, yet throughout the text the authors make the case for GOM. Maybe this is an oversight, but quite an important one considering the context of the study and does suggest the authors have had some back-andforth on which term is most appropriate.

Response: We agree there is debate regarding this issue and it still needs work. We prefer to leave as RM in the paper; however, based on the calculations it is unlikely PBM was collected by the CEM sampling the outlet of the chamber; however, it is possible that some RM be it limited was collected by the inlet CEM.

Literature: 1. Foken, T., & Wichura, B. (1996). Tools for quality assessment of surface-based flux measurements. Agricultural and forest meteorology, 78(1-2), 83-105. 2. McGillis, W. R., Edson, J. B., Ware, J. D., Dacey, J. W., Hare, J. E., Fairall, C. W., & Wanninkhof, R. (2001). Carbon dioxide flux techniques performed during GasEx-98.

Marine Chemistry, 75(4), 267-280. 3. Massman, W. J., & Lee, X. (2002). Eddy co-variance flux corrections and uncertainties in long-term studies of carbon and energy exchanges. Agricultural and Forest Meteorology, 113(1), 121-144. 4. Lindberg, S. E., Kim, K. H., Meyers, T. P., & Owens, J. G. (1995). Micrometeorological gradient approach for quantifying air/surface exchange of mercury vapor: tests over contaminated soils. Environmental science & technology, 29(1), 126-135. 5. Fritsche, J., Obrist, D., & Alewell, C. (2008). Evidence of microbial control of Hg0emissions from uncontaminated terrestrial soils. Journal of plant nutrition and soil science, 171(2), 200-209. 6. Yin, R., Feng, X., Chen, B., Zhang, J., Wang, W., & Li, X. (2015). Identifying the sources and processes of mercury in subtropical estuarine and ocean sediments using Hg isotopic composition. Environmental science & technology, 49(3), 1347-1355. 7. Eckley, C. S., Gustin, M., Lin, C. J., Li, X., & Miller, M. B. (2010). The influence of dynamic chamber design and operating parameters on calculated surface-to-air mercury fluxes. Atmospheric Environment, 44(2), 194-203.

---

## Author Comment (AC3) · 18 Feb 2019

General Comments This manuscript details mercury flux measurements made over various mining waste materials using dynamic flux chambers (DFC). The main motivation for the paper involves the use of polysulfone cation exchange membranes on the DFC sample lines as a means of collecting gaseous oxidized mercury (GOM) and thereby making estimates of GOM flux from and to the mining waste materials. After careful reading and consideration of the manuscript, I am afraid that I find the manuscript flawed and that the main conclusions of the paper are built around assumptions of potentially interfering processes that were unmeasured and/or are not as easily dismissed in reality as assumed by the authors.

[Figure]

Response: We are sorry to hear this reviewer feels this way.

Fundamentally, it is unclear to me how GOM can be emitted from these solid surface materials when nearly all studies find that GOM fluxes are dominantly in the deposition direction. The authors also do not explore in their paper a possible mechanism by which GOM emission could occur. The literature is rich in examples of how GOM is easily and quickly removed from the atmosphere to surfaces (deposited) because of its high reactivity and strong binding with surfaces. The authors point to previous work on GOM flux measurement in their introduction (lines 69-75), but even though a very small number of these studies (e.g., Skov et al. 2006) potentially found small GOM emissions, authors of previous works have attributed those emissions to measurement artifacts and/or the quick oxidation of gaseous elemental mercury emissions, making it merely appear that the fluxes are GOM. The Engle et al. (2005) paper does suggest that GOM emissions may occur with some sort of heterogeneous surface reaction.

Response: The mechanism is volatilization. Volatilization of GOM compounds from pure salts has been demonstrated in many studies (see a few refs below). In fact, we can load membranes with salts of GOM compounds by placing them above a container holding them and then verify the compound emitted using an ion chromatograph. We have added some references that point to this in the paper in the introduction.

Finley, B. D.; Jaffe, D. A.; Call, K.; Lyman, S. N.; Gustin, M., Development, testing, and deployment of an air sampling manifold for spiking elemental and oxidized mercury during RAMIX. Submitted to Environ. Sci. Technol. 2013.

Landis, M. S.; Stevens, R. K.; Schaedlich, F.; Prestbo, E. M., Development and Characterization of an Annular Denuder Methodology for the Measurement of Divalent Inorganic Reactive Gaseous Mercury in Ambient Air. Environ. Sci. Technol. 2002, 36, (13), 3000-3009.

Lyman, Seth; Jones, Colleen; O'Neil, Trevor; Allen, Tanner; Miller, Matthieu; Gustin, Mae; Pierce, Ashley; Luke, Winston; Ren, Xinrong; Kelley, Paul (2016) Automated

Calibration of Atmospheric Oxidized Mercury Measurements Environmental Science and Technology 50 12921-12927 DOI: 10.1021/acs.est.6b04211 Huang J., Miller M.B., Weiss-Penzias P., Gustin M.S. 2013 Comparison of Reactive Mercury Measurements Made with KCl-coated Denuders, Nylon Membranes, and Cation Exchange Membranes Environmental Science and Technology, 47: 7307-7316. DOI: 10.1021/acs.est.5b00098 McClure, C. D., Jaffe, D. A., Edgerton, E.S.: Evaluation of the KCl Denuder Method for Gaseous Oxidized Mercury using HgBr2 at an In-Service AM-Net Site, Environ. Sci. Technol., 48 (19), 11437-11444, 2014. Here is the text added: "The potential for GOM volatilization from surfaces has not been quantified; however, we know that GOM can be emitted from salts of a variety of GOM compounds including HgCl2, HgBr2, HgO, HgSO4, and Hg(NO3)2 (Finley et al., 2013; Landis et al., 2002; Lyman et al., 2016; Huang et al., 2013; McClure et al., 2014). Because of rapid reactions observed during the Reno Atmospheric Mercury Intercomparison Experiment (RAMIX) (Gustin et al., 2013), methods to measure GOM at a short time resolution are needed."

The authors state their findings are "a unique scientific finding" (line 363), but I feel there are too many confounding factors to confidently believe that this is likely the case. The specifics of why I would say this include: 1. As a first approximation to the suitability of this method for measurement of GOM fluxes, one would expect this method to be assessed against another relatively accepted method used in the published literature to assess fluxes. I could see this as unnecessary if all potentially uncontrolled variables were accounted for, but this is not the case in this paper.

Response: Unfortunately, there is not currently a method to compare with ours since the only alternative is using KCl denuders that have been demonstrated to be inadequate for quantifying GOM.

2. The authors rely on assumptions based on previous modelling of the DFC that flow rates are too small to induce the entrainment and potential capture of particles. This seems an overly sweeping comment to me, especially since 1) smaller particles are

than larger ones and thus the air flow rate to entrain particles would be particle size-specific and 2) there is no microscopic evidence given at all that particles were not trapped. To me, particle entrainment and capture is actually the most straightforward and most likely reason for what appeared to be, but quite possibly was not, GOM emission. In the paper, the authors have also labeled all of the GOM flux axes in their figures as "reactive mercury" (RM) fluxes, though they refer to the same fluxes in the text as GOM. As per the authors' own definition, RM includes both GOM and particulate bound mercury easily entrained.

Response: An adaptation of this response has been added to the paper. Based on the computational fluid dynamic modeling in Eckley at al. (2010), at 1 Lpm there is turbulence in the DFC, mainly near the entrances and exit. Based on the dimensions of the DFC, the friction velocity was on the order of 10-4 to 10-3 m s-1 in the flow range of 1 to 2 Lpm (Professor Jerry Lin, Lamar University-College of Engineering, Personal communication 18 February 2019). Gillette (1988) reported, in a paper that focused on trying to understand the generation of dust, threshold friction velocities for agricultural soils. Threshold friction velocity for wind erosion corresponds to the minimum wind stress needed to overcome forces holding soil particles in place. Experiments in Gillette (1988) were done using a portable wind tunnel using a variety of soils. (Dr. Heather Holmes, University of Nevada-Physics, Personal communication, 18 February 2019).

Based on this information, friction velocity for the flux chamber under operating conditions during our experiment were 2 orders of magnitude lower than those determined for a variety of soils (Gillette, 1988). In addition, as mentioned in the paper, at the on-set of the GOM flux experiments, substrates were undisturbed for ∼3 years, and were completely dry and well compacted/consolidated from previous watering experiments. Lastly, similar values were obtained for repeated experiments. Based on these pieces of information the potential for particle entrainment and suspension in the chamber was unlikely, and it is thought volatilization of GOM was occurring. Future work should work on further investigating this. Gillette, D.A., 1988. Threshold friction velocities for dust

production for agricultural soils. Journal of Geophysical Research 93, 12645–12662.